# Aging Model for Analyzing Drug-Induced Proarrhythmia Risks Using Cardiomyocytes Differentiated from Progeria-Patient-Derived Induced Pluripotent Stem Cells

**DOI:** 10.3390/ijms241511959

**Published:** 2023-07-26

**Authors:** Neil Daily, Julian Elson, Tetsuro Wakatsuki

**Affiliations:** InvivoSciences Inc., Madison, WI 53719, USA; ndaily@invivosciences.com (N.D.); julianelson@invivosciences.com (J.E.)

**Keywords:** Hutchinson–Gilford progeria syndrome (HGPS), cardiac safety, logistic regression, CiPA, cardiac safety, aging-in-a-dish

## Abstract

Among various cardiac safety concerns, proarrhythmia risks, including QT prolongation leading to Torsade de Pointes, is one of major cause for drugs being withdrawn (~45% 1975–2007). Preclinical study requires the evaluation of proarrhythmia using in silico, in vitro, and/or animal models. Considering that the primary consumers of prescription drugs are elderly patients, applications of “aging-in-a-dish” models would be appropriate for screening proarrhythmia risks. However, acquiring such models, including cardiomyocytes (CMs) derived from induced pluripotent stem cells (iPSCs), presents extensive challenges. We proposed the hypothesis that CMs differentiated from iPSCs derived from Hutchinson–Gilford progeria syndrome (HGPS, progeria) patients, an ultra-rare premature aging syndrome, can mimic the phenotypes of aging CMs. Our objective, therefore, was to examine this hypothesis by analyzing the response of 11 reference compounds utilized by the Food and Drug Administration (FDA)’s Comprehensive in vitro Proarrhythmia Assay (CiPA) using progeria and control CMs. As a sensitive surrogate marker of modulating cardiac excitation–contraction coupling, we evaluated drug-induced changes in calcium transient (CaT). We observed that the 80% CaT peak duration in the progeria CMs (0.98 ± 0.04 s) was significantly longer than that of control CMs (0.70 ± 0.05 s). Furthermore, when the progeria CMs were subjected to four doses of 11 compounds from low-, intermediate-, and high-risk categories, they demonstrated greater arrhythmia susceptibility than control cells, as shown through six-parameter CaT profile analyses. We also employed the regression analysis established by CiPA to classify the 11 reference compounds and compared proarrhythmia susceptibilities between the progeria and control CMs. This analysis revealed a greater proarrhythmia susceptibility in the progeria CMs compared to the control CMs. Interestingly, in both CMs, the compounds categorized as low risk did not exceed the safety risk threshold of 0.8. In conclusion, our study demonstrates increased proarrhythmia sensitivity in progeria CMs when tested with reference compounds. Future studies are needed to analyze underlying mechanisms and further validate our findings using a larger array of reference compounds.

## 1. Introduction

Prescription medication use is highly prevalent among individuals aged 65 and older, with approximately 90% reporting its use (Appendix A). Nearly 40% of older adults engage in polypharmacy, taking five or more drugs [1]. The United States has the largest aging population among developed countries, with 43 million people aged 65 and older in 2012, a number expected to reach 73 million by 2030 according to the U.S. Census Bureau. This increasing aging population has witnessed a 38% rise in deaths caused by heart failure (HF). By 2030, over 8 million people in the United States, approximately 1 in every 33 individuals, are projected to have HF, resulting in estimated direct medical costs of USD 53 billion [2]. HF remains the most common diagnosis among patients aged 65 and older [3]. On average, HF patients consume 6.8 prescription medications per day, amounting to a daily intake of 10.1 doses [4]. Heart disease and cancer are the primary causes of death among the elderly. The number of cancer survivors aged over 65 is increasing, comprising around two-thirds of all survivors [5]. The risk of fatal heart disease is notably higher among older, male, African American, and unmarried cancer survivors [6]. Cancer treatment itself can induce adverse cardiovascular events, leading to the emergence of the field of clinical medicine known as cardio-oncology [7].

Nearly 45% of drugs withdrawn from the market (1975–2007) were eliminated due to cardiac safety concerns [8]. The current in vitro cardiac safety FDA guidelines have been insufficient [9] in addressing cardiotoxicity issues and will be revised to require assays using cardiomyocytes (CMs) derived from human induced pluripotent stem cells (iPSCs). Despite the shortcomings of human iPSC-CMs [10], including their immaturity with poor excitation–contraction (EC) coupling [11,12], the preclinical analyses using them provide valuable information. An FDA-initiated CiPA project developed the groundwork to predict the sensitivity of predicting proarrhythmia safety using human iPSC-CMs [13].

The R&D data of products for the aging population have based their efficacy and mode of action mostly on using young animal models despite the need and market pressure to treat older people [14]. This is a critical need for the biopharmaceutical industry, including academic laboratories, to understand how aging stem cells or tissue environments respond to existing and potential treatments. The problem is that phenotypic assays using “aging-in-a-dish” methods that recapitulate human physiological and pharmaceutical responses to evaluate the efficacy and toxicity of existing and potential treatments are lacking. Therefore, developing cost-effective yet highly predictive models of aging hearts is the highest priority among various aging-in-a-dish systems. Such a system would have a medical impact on estimating the cardiac safety of drugs used to treat aging patients.

Despite aging patients consuming the majority of prescription drugs [3], this absence severely limits the efficacy of aging care and advancing new product development for treating the aging population safely and effectively. From a technical standpoint, cell culture modeling to replicate phenotypes in aging, such as senescence, has been resource-intensive, and it is challenging to consistently show the inverse relationship of donor age to proliferative life span [15]. An alternative solution to the primary cultures of aging cells is desired.

Hutchinson–Gilford progeria syndrome (HGPS, progeria) is an ultra-rare disease that prematurely exhibits multiple features of aging [16,17]. A single-base substitution, G608G(GGC > GGT), in the lamin A/C gene (*LMNA*) was discovered to produce an abnormal lamin A protein, progerin, and cause HGPS [18]. Progerin was also later found in higher passages of cultured fibroblasts in normal and aged skin [19,20,21]. A later study showed an age-related increase in progerin expression isolated from coronary artery biopsies of 29 1-month-old to 97-year-old individuals [22]. While little is known about the mechanisms, most of the cells in various organs in HGPS and a subset of senescent fibroblasts accumulate progerin in the inner nuclear membrane. We hypothesized that cardiomyocytes (CMs) derived from iPSCs reprogrammed from progeria patients recapitulate the phenotypes of aging cardiomyocytes. Therefore, we compared the proarrhythmia sensitivity of progeria and control CMs using 11 sub-selected reference compounds from the CiPA study, in which the experts selected a set of 28 compounds with well-defined cardiac electrophysiology, and known clinical characteristics were identified and categorized into high, intermediate and low risk of Torsade de Pointes (TdP) based on published reports [13]. Chloroquine was added due to its recent use as an antiviral against SARS-CoV-2 and concerns about its arrhythmogenic risk.

## 2. Results

The peak width durations of calcium transient (CaT) at 80% recovery in the progeria and control CMs were 0.98 ± 0.04 (*n* = 210) and 0.70 ± 0.05 (*n* = 315) seconds, respectively. The CaT profiles of the progeria CMs were observed to be longer than those in control CMs (Figure 1). While we did not compare the human ether-à-go-go-related gene (hERG) channel expression between the progeria and control CMs, the extension of CaT duration in the progeria CMs suggests a diminished hERG channel. In fact, the age-related diminished expression of the hERG channel was measured in non-progeria CMs, a factor known to contribute to the duration of action potentials (specifically, QT elongation due to reduced hERG activity) [23,24]. Therefore, by screening reference compounds using both control and progeria CMs, we could potentially analyze differences in susceptibility to proarrhythmia compounds.

Following several days of culturing each type of CM in a 384-well plate, they established cell–cell contacts and a steady rhythm of spontaneous cardiac contraction. After baseline calcium transient (CaT) recordings were taken, we administered escalating concentrations (six replicates for each dose) of the 11 reference compounds listed in Table 1, along with DMSO. To analyze the changes in profiles of CaTs, we compared six phenotypic parameters between the control and progeria CMs exposed to high-, intermediate-, and low-proarrhythmic-risk reference compounds, respectively. The high-risk compounds, quinidine and dofetilide, significantly increased the P80 and 8/2 of the progeria CMs (Figure 2F,G) compared to control CMs (Figure 2B,C). The increasing dose reduced the amplitudes of the control CMs and stopped beating at their highest dose. The decreases in CaT amplitudes observed in the control CMs did not generate a profile indicative of arrhythmia, such as early after-depolarization (EAD). The intermediate-risk compounds, chloroquine and chlorpromazine, displayed similar patterns of increased sensitivity in both the progeria and control CMs (Figure 3).

In general, the low-risk compounds did not significantly elongate P80 (Figure 4). However, diltiazem induced irregular beat patterns, as evidenced by an increased standard deviation of cycle lengths. Interestingly, the low-risk compounds altered the cycle lengths and their standard deviations more substantially in the control CMs compared to the progeria CMs. Radial plots also unequivocally indicated that Bepridil and cisapride exerted minimal effects on the six phenotypic parameters studied.

The selected 11 reference compounds were classified into high-, intermediate-, and low-proarrhythmic-risk categories, based on clinical data. Our experiments used the same concentration ranges as the CiPA study to treat samples (control and progeria CMs). One of the CiPA’s classifications of proarrhythmia sensitivity hinged on a logistic regression analysis with three model predictors: (1) the specific type of proarrhythmia waveform, (2) the P80 at the maximum plasma concentration (Cmax) of a drug, and (3) the maximum value of P80. We applied these model predictors of CaT to classify the compounds.

The analysis of the three model predictors demonstrated that both the progeria and control CMs reacted as anticipated based on their assigned proarrhythmia risk levels (Figure 5). High-risk compounds typically increased predictor values more significantly than those in the intermediate- and low-risk categories. Interestingly, the progeria CMs showed heightened sensitivity, with the predictor values seeing greater increases compared to the control CMs.

The response to Bepridil, in both the progeria and control CMs, was not as pronounced as with other high-risk compounds. This aligns with observations made in the CiPA study. When analyzing normalized P80@Cmax and P80_Max, it became evident that the dynamic range of the progeria CMs surpassed that of the controls. Particularly noteworthy was the enhanced proarrhythmia sensitivity of chloroquine in progeria CMs. This is intriguing, considering previous reports of chloroquine’s heightened sensitivity in COVID-19 patients aged over 65 [25], indicating the potential value of further analysis using progeria CMs.

For risk classification, we employed a three-level logistic regression based on predictor values. For intermediate-risk compounds, cisapride and chlorpromazine, the risk prediction combining intermediate- and high-risk categories surpassed a 0.8 threshold in the progeria CMs, as opposed to the control CMs (Figure 6). However, compounds in the low-risk category remained below the 0.8 threshold for both the progeria and control CMs.

In general, the proarrhythmia probability for the majority of compounds was higher when assessed using the progeria CMs. However, it is important to note that certain compounds, such as Ranolazine, demonstrated greater sensitivity in the control CMs.

## 3. Discussion

Our hypothesis suggested that the compound-induced proarrhythmia sensitivity in progeria CMs would be higher than that in control CMs. To evaluate this hypothesis, we employed 11 reference compounds and tested their effects on altering CaT profiles. Our multiparameter analysis of CaT phenotypes revealed higher sensitivity in progeria CMs to the reference compounds compared to control CMs. These differences were quantitatively assessed using multinomial regression analysis. However, despite the increased sensitivity in the progeria CMs, the low-risk compounds did not surpass the risk threshold of 0.8. This threshold value was utilized in the CiPA analysis to ensure that low-risk compounds did not exceed this level. As observed in the CiPA analysis, we also observed the lower responsiveness of Bepridil to induce proarrhythmia phenotypes in both cell types. This can be predicted to have been due to its dual effect to inhibit potassium channels and L-type calcium channels [26].

Our aging model of progeria CMs exhibited a prolonged duration of CaTs, as expected in the prolonged QT interval of aging hearts. A prolonged duration of cardiac action potential can cause life-threatening cardiac arrhythmia. Long-QT syndrome (LQTS) can be inherited or acquired through medications or medical conditions. The Center for Disease Control (CDC) conducted the US National Health and Nutrition Examination Survey (NHANES) II and III studies, gathering QT interval data from 6173 men and 7454 women aged between 25 and 90 years [27]. The analysis derived from these studies concluded that the prolongation of QTc (the QT interval corrected for heart rate) correlated with aging, with this effect being particularly noticeable in men. Consequently, elderly patients who already have prolonged QTc may be at increased risk for drug-induced arrhythmia [28]. However, further validation is necessary due to the complex abnormalities observed in the electrocardiograms of progeria patients [29,30] and mouse models [31].

Potential contributing factors that lead to heart failure in aging hearts include CM loss and hypertrophy [32] and an increasing number of fibroblasts [33]. For instance, heart failure with preserved ejection fraction initiated by activated fibroblasts is becoming an epidemic among the increasing population of aging Americans [34]. The current data were obtained using cultured iPSC-derived CMs. However, it is important to note that the adult heart [35], including the aging heart, is composed of multiple cell types such as CMs, fibroblasts, endothelial cells, and smooth muscle cells. These different cell types contribute to modulating electrophysiology in distinct ways. Therefore, it is essential to recognize the limitations of analyzing CMs alone, and future studies may require more complex systems to screen reference compounds.

To address the above limitations, we constructed engineered heart tissues (EHTs) [36] using cardiac fibroblasts and CMs derived from control and progeria iPSCs. Preliminary data from those EHTs suggest that progeria-derived EHTs exhibit longer durations of action potential, calcium transients (CaTs), and force of contraction compared to control EHTs. However, further validation is needed to confirm these observations and explore their potential underlying mechanisms.

The model predictors used in the CiPA analysis were highly effective in logistic regression analysis for classifying reference drugs based on proarrhythmic risk. However, the dose-dependent effects and irregularity of cycle length induced by the compounds were not significant contributors to the regression model predictors. The use of iPSC-CMs in drug-induced proarrhythmia assays offers advantages, such as their ability to beat spontaneously and their suitability for high-throughput dose-dependent analyses. The complexity of analyzing spontaneous beating iPSC-CMs can be mitigated by employing electrical pacing. In the analysis of the FDA group that performed the original CiPA analysis, the same logistic regression modeling was applied to optically paced iPSC-CMs to determine the proarrhythmic TdP risk category using CiPA’s 28 drugs [37]. Unfortunately, this study did not observe substantial improvement compared to iPSC-CMs without electrical pacing.

To analyze phenotypes of CaTs, we employed multiple parameters, including standard deviations of cycle lengths as an indicator of irregular beating. By including different model predictors, such as standard deviations of cycle lengths, we aimed to explore the potential for obtaining different results. Our ongoing research involves testing various model predictors that can consider different phenotypes of drug-induced proarrhythmia detection, including dose dependence. The medical impact of the results described here includes the potential effect of aging on increasing sensitivities of drug-induced arrhythmia risks. The study using a limited number of cell lines cannot discuss the extent of aging impacted on electrophysiology. Clinical data of the effect of prolonged QT duration on aging populations suggest this phenomenon deserves more attention.

The medical impact of the results suggests a potential effect of aging on increasing the sensitivities of drug-induced arrhythmia risks. Although this study used a limited number of cell lines, it still highlights the importance of considering the impact of aging on electrophysiology. Clinical data indicating prolonged QT duration in the aging population further support the notion that age-related changes in cardiac electrophysiology should receive more attention in healthcare. Understanding the potential impact of aging on cardiac electrophysiology and drug-induced arrhythmia risks can have significant clinical implications. In summary, the study’s findings, though limited in scope, emphasize the significance of investigating the impact of aging on electrophysiology and the need for further research and attention toward this phenomenon in clinical practice.

## 4. Methods

### 4.1. iPSC Culture

The iPSC lines (HGADFN167 iPS 1J, HGPS, and HGMDFN090 iPS 1C, control) were obtained from William Stanford’s laboratory from the University of Ottawa with approval from the Progeria Research Foundation. The iPSC lines used in this study were frozen iPSCs that were thawed in a 37 °C water bath for 3 min until completely thawed. The iPSCs were transferred to a sterile culture dish coated with Matrigel (Millipore Sigma, St. Louis, MO, USA) containing pre-warmed iPSC culture medium (InvivoSciences, Inc., Madison, WI, USA). The iPSCs were maintained in a humidified incubator at 37 °C and 5% CO_2_. The medium was changed every 24 h. The iPSC culture medium was prepared by mixing a basal medium, DMEM/F12 (Thermo Fisher Scientific, Waltham, MA, USA), with the appropriate supplements and essential growth factors, such as basic fibroblast growth factor (bFGF) (Gold Biotechnology, St. Louis, MO, USA).

### 4.2. iPSC Differentiation and Cardiomyocyte Purification

iPSCs were passaged in iPSC medium so that they would reach 70–80% confluency within 24–48 h. When iPSCs reached this confluency, the medium was replaced with differentiation medium supplemented with 8 µM of CHIR99021 (CHIR) (Bio-Techne, Minneapolis, MN, USA) and cultured for 24 h. On day 3, the iPSCs were treated with 5 µM of IWP-4 (Bio-techne, Minneapolis, MN, USA) in half-conditioned medium for 48 h. Differentiated iPSCs were continuously cultured with media change every 48 h until day 14, when beating was observed. On day 14, the differentiated iPSCs were replated onto Matrigel-coated flasks and subsequently treated with glucose-free DMEM (Thermo Fisher Scientific, Waltham, MA, USA), with 5 mM lactate (Millipore Sigma, St. Louis, MO, USA). Differentiated iPSCs were cultured in this purification medium for 72 h, after which, they were cultured in Advanced RPMI (Thermo Fisher Scientific, Waltham, MA, USA).

### 4.3. Compounds

Table 1 includes 11 reference compounds (Cayman Chemical Company, Ann Arbor MI, USA) selected from the FDA-initiated CiPA project [13], which were categorized as having high, intermediate, and low proarrhythmic risks based on clinical data. We used the same concentration ranges to treat samples (six and four wells for each concentration of each compound using control and progeria CMs, respectively).

**Table 1 ijms-24-11959-t001:** Reference compounds were selected from the compound list of CiPA project.

Drug	Category	Cmax [nM]	Dose1 [nM]	Dose2 [nM]	Dose3 [nM]	Dose4 [nM]
Dofetilide	High	2.14	0.3	1	3	10
Bepridil	High	31.5	35	105	210	315
Quinidine	High	843	100	300	900	2700
Terfenadine	Mid	0.286	0.08	0.8	8	80
Chloroquine	Mid	250	250	750	2500	25,000
Chlorpromazine	Mid	34.5	35	350	1050	3500
Cisapride	Mid	2.58	2.5	7.5	25	125
Diltiazem	Low	128	3	10	30	90
Mexiletine	Low	2500	625	1250	2500	3750
Ranolazine	Low	1948	1000	2300	6900	15,000
Verapamil	Low	45	1	10	50	150

### 4.4. Assay

Multi-parameter phenotyping: We established the multi-parameter phenotyping of calcium transients (CaTs). After automatically finding all the CaT cycles using iVSurfer (InvivoSciences, Inc., Madison, WI, USA), the six parameters, including peak width duration 80 (P80), peak width duration 80/20 (8/2), cycle length (CL), amplitude (AMP), the number of peaks in the recorded timeframe (PN), and the standard deviation of cycle lengths of all peaks (CSD) were computed (Appendix A).

### 4.5. Radial Plots

The six parameters were calculated as a percent change from the baseline (before compound addition). A sigmoid function (exp(−9(*x* − 3) + 1)^−1^ was applied to 8/2 to improve indication of early afterdepolarization (EAD). CL was equal to the duration between two peaks. PN was the number of calcium transient cycles found within the recording. Standard deviations of CL values, CSD, represented the variance in CL, indicating instable CaT wave trains. The log-modulus transformation, *L*(*x*) = *sign*(*x*)log (|*x*| + 1), was applied to visualize a wide range of parameter values. Plotting dose-dependent relative changes in six parameters in radial plots, the trends in compound-induced changes to parameters were visualized to compare similarities and differences of compound-induced responses.

### 4.6. Logistic Regression

Logistic regression was performed using Scikit-learn, the open-source Python library for machine learning. The logistic regression function was used with the ‘multinomial’ class for multiple features (multi_class = ‘multinomial’) because there were three drug categorizations (high, intermediate, and low). The LBFGS solver (solver = ‘lbfgs’) was used with 1000 maximum iterations (max_iter = 1000) as the maximum number of iterations taken for the solver to converge. The logistic regression model was created and trained, with x_train equaling the parameter values of the calcium transients and y_train equaling the drug categorization according to CiPA (i.e., high, intermediate, low). The predicted probabilities were calculated for each drug and plotted in a stacked bar chart.

model = LogisticRegression(multi_class = ‘multinomial’, solver = ‘lbfgs’, random_state = 0, max_iter = 1000)result = model.fit(x_train,y_train)probabilities = model.predict_proba(drug)

## 5. Conclusions

The present study aimed to compare the phenotypic responses of progeria and control iPSC-derived CMs to 11 reference compounds associated with proarrhythmia. Our observations revealed the increased sensitivity of progeria CMs to the proarrhythmic effects of reference compounds compared to control CMs. However, it is important to note that the low-risk compounds tested did not exceed the risk threshold of 0.8 utilized in the CiPA analysis.

Considering the natural elongation of QT intervals in aging populations and their frequent consumption of multiple drugs, the CM model derived from progeria iPSCs can be a useful indicator for predicting the proarrhythmia risk associated with prescription drugs. As the aging population constitutes a significant consumer of prescription drugs, it is important to consider the use of an aging model for predicting the risk of proarrhythmia. It is worth noting that further research and validation are required to confirm these findings and to establish the broader applicability and reliability of using the progeria CM model as a predictive tool for proarrhythmia risk assessment induced by prescription drugs.

## Figures and Tables

**Figure 1 ijms-24-11959-f001:**
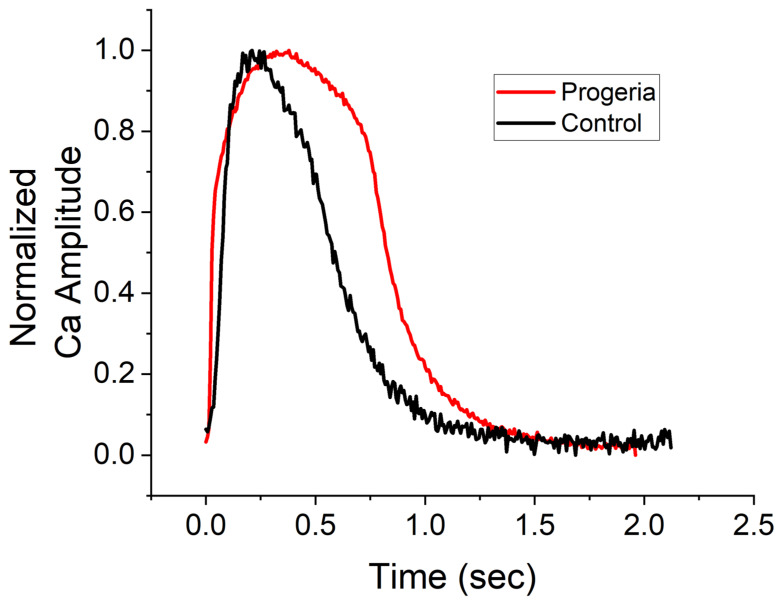
Profiles of calcium transients. Changes in calcium concentrations over time of control (black line) and progeria (red line) cardiomyocytes derived from their iPSC lines. Profiles were normalized their peak.

**Figure 2 ijms-24-11959-f002:**
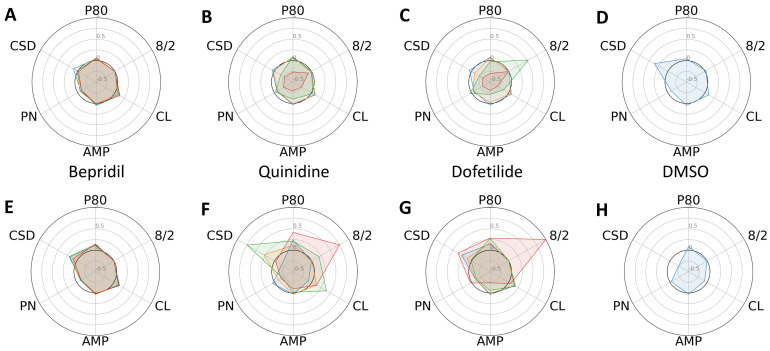
Dose-dependent changes in phenotypic parameters in response to high-risk compounds. Radial plots of control (**A**–**D**) and progeria (**E**–**H**) in response to Bepridil (**A**,**E**), quinidine (**B**,**F**), dofetilide (**C**,**G**), and DMSO (**D**,**H**) showed dose-dependent changes. Each parameter was normalized by that of before compound addition. Doses 1–4 are shown in blue, orange, green, and red.

**Figure 3 ijms-24-11959-f003:**
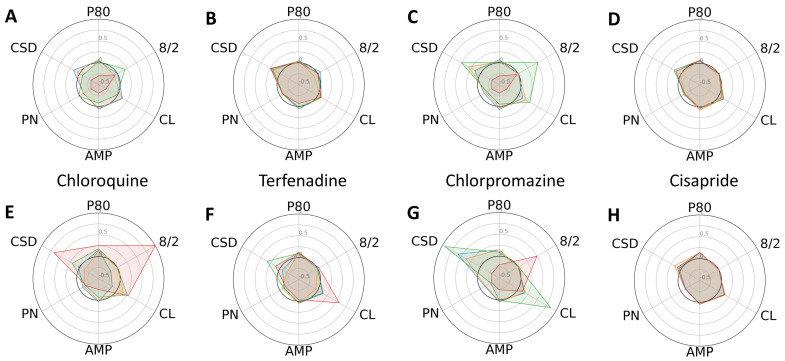
Dose-dependent changes in phenotypic parameters in response to intermediate-risk compounds. Radial plots of control (**A**–**D**) and progeria (**E**–**H**) in response to chloroquine (**A**,**E**), Terfenadine (**B**,**F**), chlorpromazine (**C**,**G**), and cisapride (**D**,**H**) showed dose-dependent changes. Each parameter was normalized by that of before compound addition. Doses 1–4 are shown in blue, orange, green, and red.

**Figure 4 ijms-24-11959-f004:**
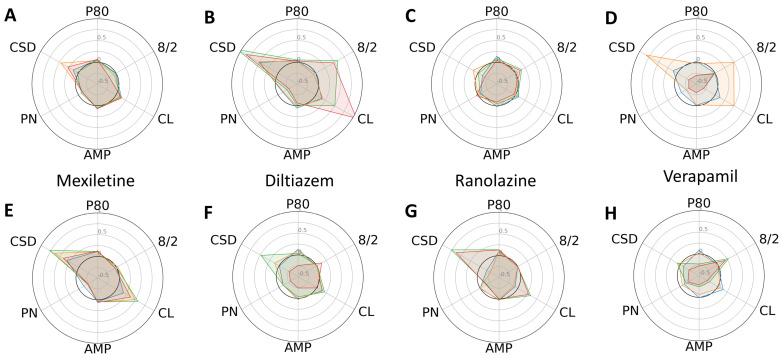
Dose-dependent changes in phenotypic parameters in response to low-risk compounds. Radial plots of control (**A**–**D**) and progeria (**E**–**H**) in response to Mexiletine (**A**,**E**), diltiazem (**B**,**F**), Ranolazine (**C**,**G**), and Verapamil (**D**,**H**) showed dose-dependent changes. Each parameter was normalized by that of before compound addition. Doses 1–4 are shown in blue, orange, green, and red.

**Figure 5 ijms-24-11959-f005:**
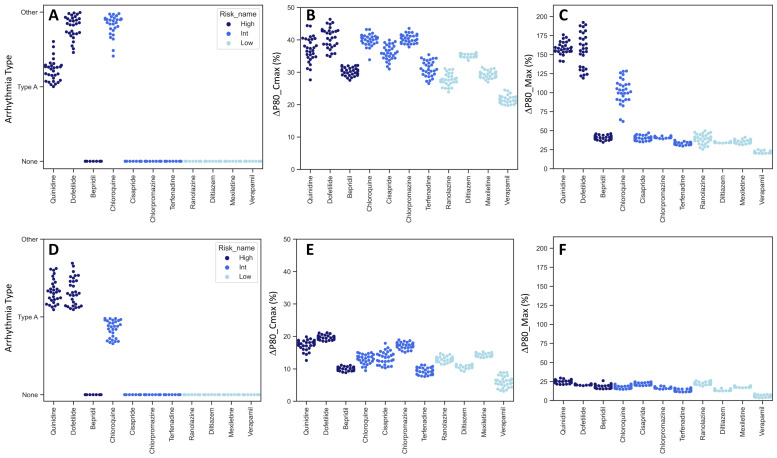
Model predictors of compound classification. The values of three model predictors used for regression analysis were compared between progeria (**A**–**C**) and control (**D**–**F**) CMs. CaT profiles are categorized into three arrhythmia types, None (no change), Type A (EAD-like), and Other (extended duration and beat secession) (**A**,**D**). Percent change of P80 at Cmax (**B**,**E**). Percent change of maximum P80 (**C**,**F**). Dark blue, blue, and light blue correspond to high-, intermediate-, and low-risk compounds, respectively.

**Figure 6 ijms-24-11959-f006:**
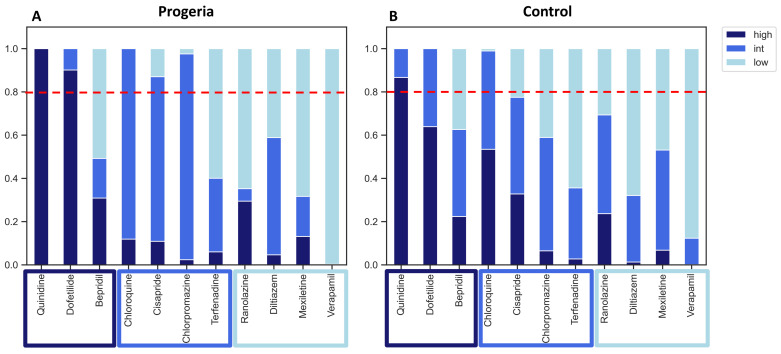
Regression analysis of 11 compounds. Proarrhythmia risk probabilities (scale 0–1) of progeria (**A**) and control (**B**) CMs against 11 reference compounds were analyzed. Multi-nominal regression of high-, intermediate-, and low-risk categories were shown in dark blue, blue, and light blue, respectively. A horizontal dotted red line showed a threshold of 0.8 probability of arrhythmia risk.

## Data Availability

Not applicable.

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
