# Peer review of "Aging Model for Analyzing Drug-Induced Proarrhythmia Risks Using Cardiomyocytes Differentiated from Progeria-Patient-Derived Induced Pluripotent Stem Cells"

_ijms, 2023, doi:10.3390/ijms241511959_

Round 1

Reviewer 1 Report

The submitted manuscript is the study of a new experimental model to be used to evaluate safety of drugs and proarrythmia  risk associated with their administration in the elderly. 

The manuscript proposes a good model. My concern is about limited availability of HGPS patient-derived cells.

I propose the following amendments:

1. progeria fibroblasts and nearly all HGPS cells (except those in the brain) express progerin, but senescent fibroblasts do not always express progerin. Please, mention only HGPS cells in the last paragraph of the introduction section.

2. In the first paragraph of the results "extension of CaT duration in progeria CMs is consistent with age-related diminished expression of the hERG channel" suggests that hERG channel has been studied in HGPS, which is not the case. Please, rephrase.

3. In the methods section, please replace "stem cells" with "induced Pluripotent stem cells", if you are referring to iPSC.

4. Please, add in the introduction and discussion the rationale behind the choice of tested drugs and the impact on the health system.

The English language is OK, I only found minor typing errors.

Author Response

Reviewer #1.

“The manuscript proposes a good model. My concern is about limited availability of HGPS patient-derived cells.”

We provided sources and specific lines of HGPS patient and control iPSCs. Various versions protocols used for deriving cardiomyocytes are available from public domain.   

“1. Progeria fibroblasts and nearly all HGPS cells (except those in the brain) express progerin, but senescent fibroblasts do not always express progerin. Please, mention only HGPS cells in the last paragraph of the introduction section.”

We appreciate the reviewer for the clarification. I apologize for the initial oversight regarding the fibroblast study. It appears that the study by Dr. Collins' laboratory at the NIH did not find progerin in primary fibroblasts from skin biopsies in early passages and observed that progerin expression was associated with the passage number rather than donor age.

However, newer data demonstrated an age-related increase in progerin expression in coronary artery biopsies obtained from 29 individuals ranging from 1 month to 97 years old. The majority of progerin-positive cells were isolated from the adventitia layer and were negative for smooth muscle actin, suggesting that they were vascular fibroblasts.

These findings suggest that there may be differences in progerin expression between fibroblasts derived from skin biopsies and those isolated from coronary artery biopsies. It highlights the importance of considering tissue-specific effects and the potential involvement of vascular fibroblasts in progerin accumulation in the arterial walls with age.

The manuscript was edited accordingly.

“2. In the first paragraph of the results “extension of CaT duration in progeria CMs is consistent with age-related diminished expression of the hERG channel” suggests that hERG channel has been studied in HGPS, which is not the case. Please, rephrase.”

Thank you for the comment. To clarify, while your study documented the age-related diminished expression of the hERG channel, it did not specifically measure hERG channel expression in progeria cardiac myocytes (CMs) compared to control CMs.

The sentence try to describe that the extension of CaT (calcium transient) duration observed in progeria CMs could potentially be explained by the diminished expression of the hERG channel. This association implies that the age-related phenotype seen in HGPS (Hutchinson-Gilford Progeria Syndrome) may also be observed in aging hearts, where similar changes in hERG channel expression and CaT duration might occur.

The manuscript was revised accordingly.

“3. In the methods section, please replace “stem cells” with “induced pluripotent stem cells”, if you are referring to iPSC.”

We have updated the methods section by replacing “stem cells” with “iPSCs”.

“4. Please, add in the introduction and discussion the rationale behind the choice of tested drugs and the impact on the health system.”

The 11 reference compounds were chosen from a list of the FDA’s Comprehensive in vitro Proarrhythmia Assay (CiPA). The FDA, academia, and industry consortium selected and categorized those compounds based on their known torsadogenic risks. Chloroquine was added due to its recent use as an antiviral against SARS-CoV-2 and its concerns about arrhythmogenic risk.

Choice of compounds has an impact on evaluating potential torsadogenic risks in new drugs. The FDA requires addition of such data as part of preclinical study package.  

Reviewer 2 Report

In this study, the Authors evaluated the proarrythmia risks associated with drug use, particularly in the context of elderly patients who are the primary consumers of prescription drugs. The authors propose the use of a model called "aging-in-a-dish," which involves the differentiation of cardiomyocytes (CMs) from induced pluripotent stem cells (iPSCs) derived from Hutchinson-Gilford progeria syndrome (HGPS) patients. The objective of the study was to analyze the response of 11 reference compounds, commonly utilized in the FDA's Comprehensive in vitro Proarrhythmia Assay (CiPA), using both progeria and control CMs.  The Authors demonstrated an increased proarrhythmia sensitivity in progeria CMs when exposed to the reference compounds. Further studies are required to explore the underlying mechanisms and validate these findings using a larger array of reference compounds.

This study contributes to our understanding of proarrhythmia risks associated with drug use, especially in the context of aging and the challenges faced in acquiring appropriate models for preclinical studies. The use of iPSCs derived from HGPS patients provides a unique opportunity to mimic the phenotypes of aging CMs and investigate drug-induced changes in cardiac function. This study can be published. There are only several small concerns:

1 Abstract: Please, provide a short definition of “Proarrythmia”

2 Please, outline medical applications of the model.

Results:

3 Please match the value range and Y-axis scale in Fig. 5, E and F.

Discussion:

In conclusion – please insert the explanation of how your results can be applied in medicine.  

English is very good,

There are only several small misprints

Author Response

Reviewer #2

We appreciate the Reviewer #2’s valuable comments, and our responses are summarized below. We revised the manuscript accordingly.

Abstract:

1 “Please, provide a short definition of “Proarrhythmia”

A definition of Proarrhythmia was added to the abstract.

2 “Please, outline medical applications of the model”

The medical applications of “aging-in-a-dish” model include analysis of drug-induced proarrhythia risk during drug development and regulatory approval in preclinical stage.

Results:

3 “Please match the value range and Y-axis scale in Fig. 5, E and F.”

The value ranges of Y-axis in Fig. 5E, F are matched with those in Fig. 5B,C in order to emphasize the increased sensitivity of progeria CMs.     

Discussion:

“Please insert the explanation of how your results can be applied in medicine.”

We added a short section describing potential impact of our results in medicine.

Reviewers 1 & 2:

Minor editing of English language required:

Edit of the spelling of proarrhythmia in the Abstract was fixed.

Edit of the sentence containing “an ultra-rare premature aging syndrome” in Abstract was fixed.

Other errors were fixed.